# Engineering ultra-strong electron-phonon coupling and nonclassical electron transport in crystalline gold with nanoscale interfaces

Shreya Kumbhakar [1,5] ✉, Tuhin Kumar Maji[1,5] ✉, Binita Tongbram[1], Shinjan Mandal [1], Shri Hari Soundararaj[1,2], Banashree Debnath [1], Phanindra Sai T [1], Manish Jain [1], H. R. Krishnamurthy [1,3], Anshu Pandey [4] & Arindam Ghosh [1] ✉

Electrical resistivity in good metals, particularly noble metals such as gold (Au), silver (Ag), or copper, increases linearly with temperature ($T$) for $T > \Theta_D$, where $\Theta_D$ is the Debye temperature. This is because the coupling ($\lambda$) between the electrons and the lattice vibrations, or phonons, in these metals is weak, with $\lambda$ ~ 0.1–0.2. In this work, we outline a nanostructuring strategy of crystalline Au where this concept of metallic transport breaks down. We show that by embedding a distributed network of ultra-small Ag nanoparticles (AgNPs) of radius ~ 1–2 nm inside a crystalline Au shell, the electron-phonon interaction can be enhanced, with an effective $\lambda$ as high as $\approx 20$. With increasing AgNP density, the electrical resistivity deviates from $T$-linearity and approaches a saturation to the Mott-Ioffe-Regel scale $\rho_{MIR} \sim ha/e^2$ for both disorder ($T \to 0$) and phonon ($T \gg \Theta_D$)-dependent components of resistivity (here, $a = 0.3$ nm, is the lattice constant of Au).

The phenomenon of resistivity saturation in disordered metals, especially transition metals and compounds that are usually good superconductors at low temperatures, has remained an open problem in solid-state physics for over 50 years[1–9]. It is commonly agreed that saturation occurs when the scattering of electrons by either defects or thermal excitations (phonons) shrinks the mean free path to the order of inter-atomic spacing - the so-called Mott-Ioffe-Regel (MIR) limit[4,10]. The quasi-particles become incoherent, resulting in a class of 'bad metals'. Although a comprehensive theoretical understanding of the saturation remains elusive[6–9], the electron-phonon coupling (EPC) seems to play a key role, and resistivity saturation has been linked to, for example, the breakdown of Born-Oppenheimer approximation and delocalization of electronic states[11], intermediate coupling of phonons to local electronic levels or hopping integrals[6,7] or phonon-driven parallel channels of electrical conduction[8,9]. In fact, the limit of extremely strong EPC remains poorly understood in the context of metallic

transport in general, at least experimentally, and it is not clear though whether the scattering rate of electrons would be limited by, for example, possible universal (Planckian) bound[12,13], polaronic deformation[9,14], or indeed, the stability of the metallic state itself against polaronic self-trapping[15]. The lack of understanding is partly caused by the fact that in most naturally occurring or synthesized metallic solids/alloys so far, the EPC parameter ($\lambda$) has not been found to exceed ~2[16], let alone a systematic tunability of $\lambda$ over a broad range in the same system. Controlled incorporation of disorder has been shown to assist saturation of resistivity at high temperatures, leading to the so-called 'Mooij correlation'[17], but the EPC remains largely intrinsic and $\lesssim 1$, and the fate of metallic transport in the limit $\lambda \gg 1$ remains experimentally unknown.

Well-known methods to engineer EPC in solids often depend on confinement or localization of both electrons and phonons, enabled by defects, topological disorder or interfaces that cause acoustic

[1]Department of Physics, Indian Institute of Science, Bangalore, India. [2]Materials Science and Engineering, University of California Riverside, Riverside, CA, USA. [3]International Centre for Theoretical Sciences, Tata Institute of Fundamental Research, Bangalore, India. [4]Solid State and Structural Chemistry Unit, Indian Institute of Science, Bangalore, India. [5]These authors contributed equally: Shreya Kumbhakar, Tuhin Kumar Maji. ✉e-mail: shreyak@iisc.ac.in; tuhinmaji@iisc.ac.in; arindam@iisc.ac.in

impedance mismatch, lifting of structural symmetry, etc[18–27]. In semi-conductor quantum dots and wells, Fröhlich interaction between the charge and the electric field from the optical phonons is naturally enhanced when the chemical bonds are polar in nature[28], but placing an interface, for example, an antiphase boundary, was shown to increase the Huang-Rhys factor (a measure of the EPC in optically excited semiconductors) by orders of magnitude even in nominally weakly polar III-V crystals[18]. In metals, however, attempts to increase EPC by confining electrons[19], confining phonons[20], interfacial charge transfer[21], enhanced electron surface scattering[22,29], optical driving[23], or application of stress[24–27] resulted only in a moderate increase in $\lambda$ within a factor of ~ two. An alternate strategy involves core@shell (e.g., Au@Ag or Au@Ag@Pt) nanostructures, where the effective EPC can be continuously tuned with core/shell mass fraction owing to sound velocity mismatch at the hetero-interface[30]. Charge scattering at such heterointerfaces seems to determine the residual resistivity at low temperatures ($T$) in Ag@Au core@shell nanostructures[31], but their effect on the EPC has not been investigated.

In this work, we have investigated EPC in noble metal hybrids consisting of a network of nanometre-sized Ag cores embedded in a crystalline Au matrix. The small intrinsic EPC and near-identical lattice constants of Ag and Au that preserve a global translation symmetry, provide a simple platform for analyzing the metallic state resistivity. Using electrical transport and point contact spectroscopy, we find that both static disorder and the EPC increase dramatically with increasing density of Ag nanoparticles (AgNP, core), *i.e.*, the proliferation of buried Ag-Au interfaces. At intermediate volume fractions of Ag, $\lambda$ as

high as ~20 could be observed, over ten times that of any known metal. This regime is also associated with a strong saturation in electrical resistivity that could be monitored by varying the EPC over nearly two orders of magnitude for the first time.

Figure 1a–d show the high-resolution transmission electron microscopy (HRTEM, Fig. 1a–b; TEM Fig. 1c) and scanning electron microscopy (SEM, Fig. 1d) image of the hybrid at increasing length scales. The building block consists of solution-processed ~20–30 nm Au shells, each of which encloses multiple AgNPs of ~2 to 5 nm diameter (Fig. 1a,b). The shells are subsequently fused, or 'cross-linked', and compacted to form macroscopic films on a glass substrate with pre-patterned electrical leads (Fig. 1e, Fig. S6 in Supplementary Information). The Methods and Supplementary Information sections I–III describe the chemical synthesis, characterization, and film-making processes in detail. Figure 1a (and also Fig. S3 in Supplementary Information) emphasizes the sharp interface between AgNPs and the Au shell, which was found to be the case irrespective of the density ($\approx F/r_{Ag}^3$) of the AgNPs (here, $r_{Ag}$ and $F = V_{Ag}/(V_{Ag} + V_{Au})$ are the radius of the AgNP and the net relative volume fraction of Ag in the hybrid, respectively). Typically, we synthesize AgNPs of $r_{Ag} \approx 1–2$ nm and vary $F$ to tune the concentration of AgNPs, and thereby the inter-AgNP distance, $d_{Ag} = 2r_{Ag}/F^{1/3}$, and overall interface density, $F/r_{Ag}$.

Figure 1f shows the $T$-dependence of the electrical resistivity ($\rho$) for Ag@Au nanohybrid films of different $F$ between 6 K and 300 K. All the films showed metallic behaviour where $\rho$ decreases monotonically with decreasing $T$ with very little or no evidence of upturn even at $T$ ~ 0.3 K (Fig. S7 in Supplementary Information). Thorough

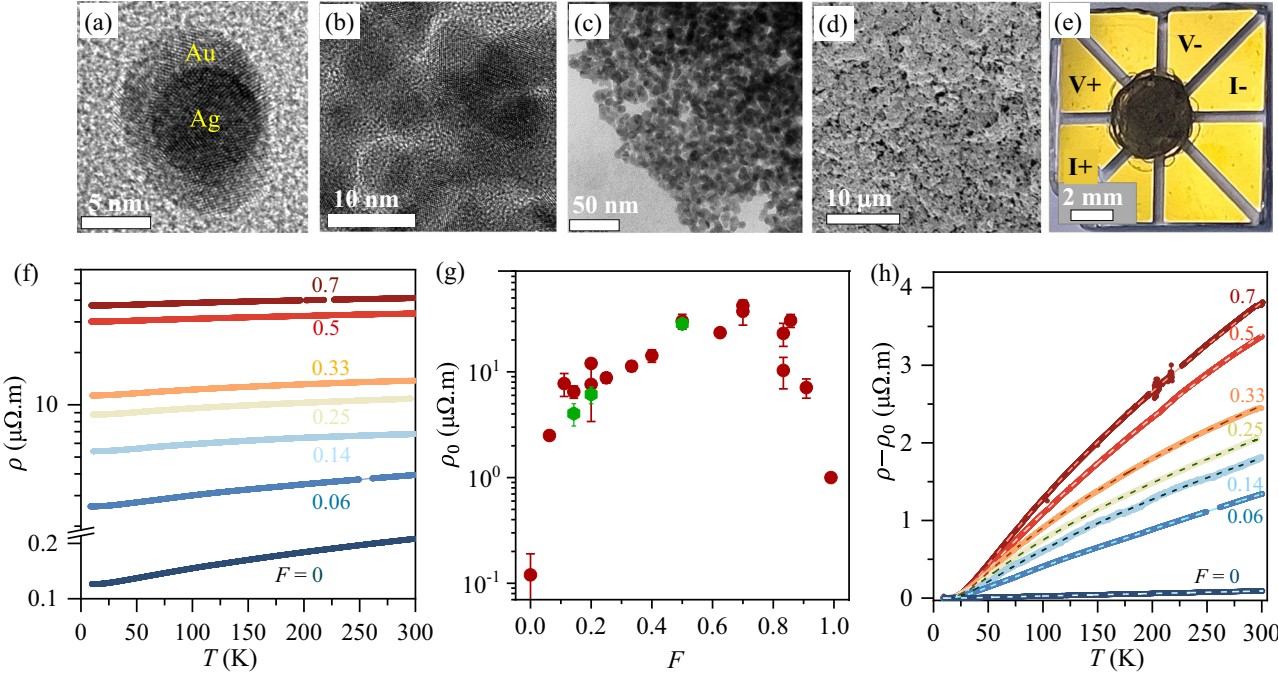

**Fig. 1 | Structure and electrical transport. a–e** Ag@Au nanohybrids at different length scales. **a** High-Resolution Transmission Electron Microscopy (HRTEM) image of a single AgNP of high crystallinity is embedded inside a crystalline matrix of Au. The AgNPs are typically spherical, and the interface between AgNPs and the Au host is sharp even at the atomic scale. **b** The interface sharpness remains robust even after forming a dense AgNP network within Au. **c** A snapshot of the intermediate stage of fusing (or cross-linking) of the Ag@Au nanohybrids that eventually form a continuous and compact network upon multiple loading via drop cast. **d** Scanning Electron Microscope (SEM) image of an Ag@Au film after multiple (about ten) iterations of dropcast and cross-linking. **e** Optical image of a typical Ag@Au film fabricated on pre-patterned Van der Pauw leads. $I +$ , $I -$ and $V +$ , $V -$ represent the current and volatge contacts for four-probe resistivity

($\rho$) measurements, respectively. **f** Variation of $\rho$ with temperature ($T$) for films with different Ag volume fraction $F$ (equivalent to AgNP density), showing metallic transport down to the lowest temperature. **g** Residual resistivity ($\rho_0$), defined as the value of $\rho$ observed at $T$ ~ 6 K as a function of $F$. The red and green points were evaluated from films in Van der Pauw and Hall bar geometries, respectively. (See Fig. S6 in Supplementary Information). Error bars represent the standard deviation of the channel-to-channel statistics of resistivity in the same film. **h** Variation of the $T$-dependent component of $\rho$, obtained by subtracting $\rho_0$ from $\rho$, reveals the emergence of sub-linear behaviour in $\rho$ at high temperatures with increasing $F$. The pure AuNP film is represented by $F = 0$. Dashed lines represent fit to the data using the two-component parallel channel model given by Eq. [(2)].

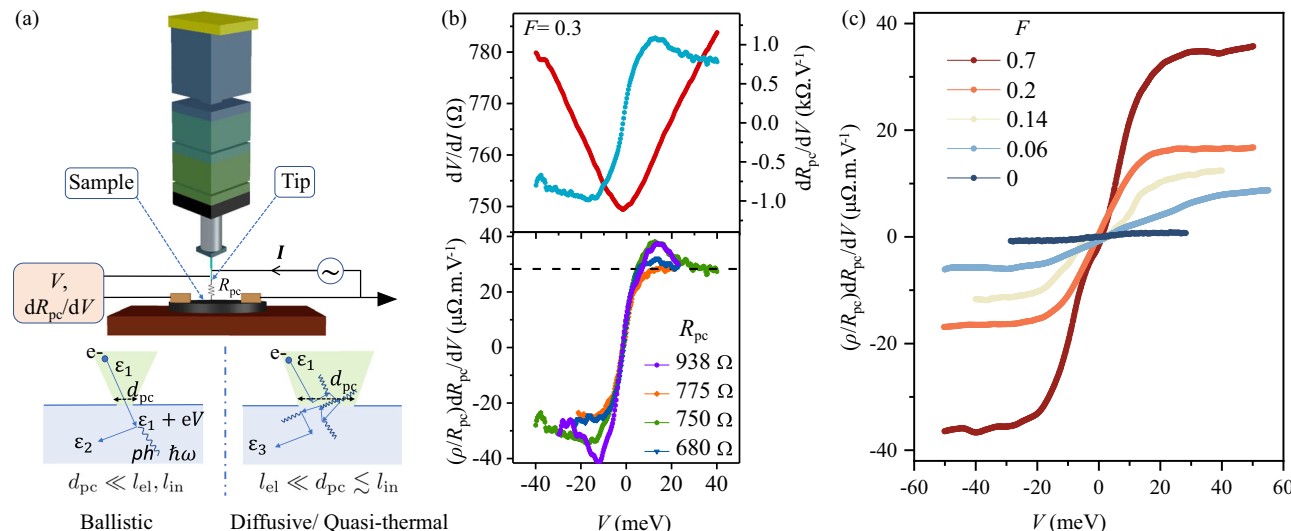

**Fig. 2 | Point contact spectroscopy of Ag@Au hybrid films. a** Schematic and processes: The tip is biased with a voltage ($V$) while the sample is grounded thereby driving current ($I$) across the tip-sample nanocontact of dimension $d_{pc}$ and resistance $R_{pc}$. An electron with initial energy $\epsilon_1$ gains an additional energy $eV$ while passing from the tip to the sample for a ballistic contact where $l_{in} \gg d_{pc}$, $l_{in}$ being the inelastic mean free path of the electrons. Straight lines represent electrons, while curly lines indicate the emission of phonons responsible for the dissipation of the excess energy. The dissipation of this excess energy by scattering in a ballistic contact leads to non-linearities in $I - V$ characteristics, which can be shown to correspond to the electron-phonon interaction (EPI) function. In the case of a quasi-thermal contact, since $l_{el}$, $\sqrt{l_{el}l_{in}/3} \ll d_{pc} \lesssim l_{in}$ the energy of the electrons is dissipated at the orifice via inelastic collisions generating non-equilibrium phonons

($l_{el}$ is the elastic mean free path). In such cases, the point contact spectrum at $eV \gg \hbar\omega_D$, is a cumulative contribution from the phonons available at all energies below $\hbar\omega_D$. (See Supplementary Information Section VII for a detailed discussion.) **b** Top panel shows the bias dependence of measured $dV/dI$ (i.e., $R_{pc}$) and $dR_{pc}/dV$ for a typical film with Ag volume fraction $F = 0.3$. The bottom panel shows the bias dependence of normalized point contact spectrum i.e. $(\rho/R_{pc})dR_{pc}/dV$ for different point contact resistances which converge to a geometry-independent background (black dashed line). **c** Bias dependence of $(\rho/R_{pc})dR_{pc}/dV$ for films with different $F$ measured at $T \approx 5$ K demonstrating an increasing background value with increasing $F$. For the pure AuNP film ($F = 0$), we have plotted $d_{pc}dR_{pc}/dV$ as discussed in Supplementary Information Section VII.D.

compaction and crosslinking result in geometric uniformity and electrical homogeneity better than ~20% (Fig. S6 in Supplementary Information) and low background resistivity $\rho$ ~0.2 $\mu\Omega$. m obtained in identically prepared films of bare Au nanoparticle (i.e., $F = 0$). The incorporation of AgNPs causes $\rho$ to increase rapidly, which decreases again when the composite becomes Ag-rich ($F \to 1$). This is seen in the variation in the residual resistivity $\rho_0$ (defined as $\rho$ at $T \approx 6$ K) with $F$, shown in Fig. 1g. Remarkably, in the intermediate range of $F$ ~ 0.4–0.8, $\rho_0$ is nearly constant and pinned to the magnitude of ~30–40 $\mu\Omega$. m, which is scale of the Mott-Ioffe-Regel limit[10], $\rho_{MIR} = 3\pi^2\hbar a/e^2 \approx 10 \mu\Omega$. $m$ of metallic resistance for Au ($a = 0.3$ nm, is the lattice constant). At low $F$ ($\lesssim 0.4$), $\rho_0$ increases linearly with the overall Ag-Au interface per unit volume, suggesting that the scattering of the electrons occurs dominantly at the buried Ag-Au interfaces (Fig. S8 in Supplementary Information)[31].

The $T$-dependent component of electrical resistivity, i.e., $\rho - \rho_0$, separately shown in Fig. 1h, contains two important features. First, the increase in the overall magnitude of $\rho - \rho_0$ with increasing $F$ at any given $T$, implies increasing contribution from phonons to resistivity, and thus enhancement in the 'effective' EPC. Secondly, the incorporation of the AgNPs also makes the $T$-dependence of $\rho$ increasingly sublinear at high temperatures for $T > \Theta_D$, where $\Theta_D$ ~ 150 K is the Debye temperature of Au. The coexistence of sub-linear $T$-dependence of $\rho$ in disordered metals with large EPC, such as the A15 compounds, has been known for many years[2,4], but it is not expected in crystalline noble metals such as Au or Ag (or their alloys). The sublinearity makes the Bloch-Grüneisen formula ($\rho_{BG}(T)$) for metallic resistivity, which derives $\rho \propto T$ for $T > \Theta_D$ by treating the EPC perturbatively, evidently inadequate except for $F = 0$ and 1 (Fig. S10 in Supplementary Information). This necessitates an alternative experimental tool to quantify the EPC parameter $\lambda$ in this case.

To estimate $\lambda$ independently, we have performed point contact spectroscopy on the Ag@Au nanohybrid films. Figure 2a schematically explains the experimental arrangement and the underlying physical processes. In order to determine the dominant transport mechanism at the point contact, we estimated the relative spatial scales of the point contact $d_{pc}$ ($\approx 10 - 30$ nm), as well as the elastic ($l_{el}$ ~1 nm) and inelastic ($l_{in}$~20 nm) lengths, where the latter was estimated from quantum transport measurements (Supplementary Information Section VII). We find $l_{el}$, $\sqrt{l_{el}l_{in}} \ll d_{pc} \lesssim l_{in}$, suggesting a regime intermediate to the diffusive and thermal transport, where the inelastic scattering events at the orifice can stimulate the generation of non-equilibrium phonons, which get trapped and increase the local temperature[32,33]. In this *quasi-thermal* regime, the spectral information is lost, but the cumulative effect of phonons at all available energies below $\hbar\omega_D$ causes a finite energy-independent 'background' that is directly proportional to the EPC. (See Methods, and Supplementary Information Section VII for more detail). $\lambda$ is then quantitatively estimated from the energy-derivative of the resistance $R_{pc}$ of a nanoscale contact between the film and a metallic Pt/Rh tip as (see derivation in Methods)[33],

$$\lambda \approx \frac{3\pi neh}{16m} d_{pc} \left[\frac{dR_{pc}}{dV}\right]_{V\to\infty} \quad (1)$$

where $V$, $n$ and $m$ are the tip-sample bias, electron density in Au and electronic mass, respectively, and $d_{pc} = \rho/R_{pc}$ is the contact diameter (Maxwell regime). The upper panel of Fig. 2b shows typical $V$-dependence of $R_{pc}$ and $dR_{pc}/dV$. At large $V$, i.e. $e|V| \gg \epsilon_t$, where $\epsilon_t$ ~ $k_B\Theta_D$ ~ 10 − 20 meV is the energy scale beyond which the Migdal-Eliashberg function (Eq. (12)) $\alpha^2\mathcal{F}(\omega) \to 0$, $[d_{pc}dR_{pc}/dV]$ becomes independent of the geometric details of the contact, as seen from the

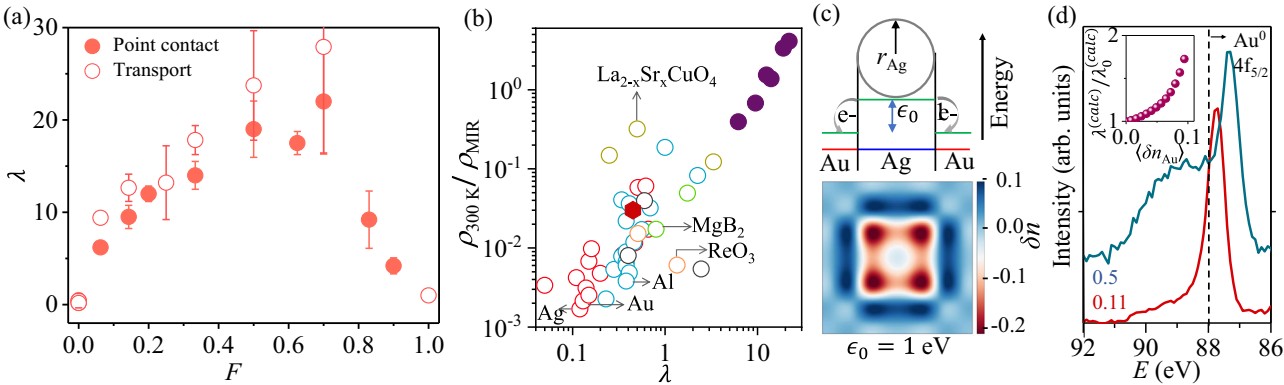

**Fig. 3 | Electron-phonon coupling in Ag@Au hybrids. a** Electron-phonon coupling constant ($\lambda$) estimated from point contact measurements and electrical transport ($\rho - T$) data as function of Ag filling $F$. Error bars in the open and filled points, representing $\lambda$ have been estimated from the error in resistivity, $\rho$, shown in Fig. 1g by error propagation. **b** Room temperature resistivity ($\rho_{300\,K}$) for different materials is normalized by the respective Mott-Ioffe-Regel resistivity ($\rho_{MIR}$) and plotted as a function of electron-phonon coupling constant, $\lambda$. Red, blue, green, and yellow open circles represent non-superconducting metals, metals/alloys that superconduct at low $T$, intermetallic compounds, and high-$T_c$ cuprates, respectively (see Supplementary Information VIII for detail). Filled red and purple circles represent films of pure Au nanoparticle and Ag@Au hybrids for different $F$ values, respectively. **c** (Top Panel): A schematic of the electrochemical potential of electrons at the Ag and Au sites across the Ag@Au nanohybrid, $\epsilon_0$ being the potential difference between them. Electrons transfer from a higher onsite potential at Ag to a lower potential in Au. (Bottom Panel): Theoretical computation of the excess electron occupancy, $\delta n$ in a square lattice toy model (See Methods and Supplementary Information Section IX), where Ag is embedded inside Au. **d** Binding energy peak of the Au-$4f_{5/2}$ peak from X-ray photoelectron spectroscopy (XPS) is shown for two different values of $F = 0.11, 0.5$. The data is shifted vertically for clarity. (See Fig. S5 and Section II.B in Supplementary Information for more details). The dotted line represents the binding energy peak of Au-$4f_{5/2}$ core-level of $Au^0$. Inset shows the theoretically computed EPC on the square lattice toy model, $\lambda^{(calc)}$ with the average excess electron occupancy ($\langle \delta n_{Au} \rangle$) on Au.

convergence of the traces at different $R_{pc}$ (lower panel of Fig. 2b). Figure 2c shows the $V$-dependence of $d_{pc}dR_{pc}/dV$ for different values of $F$. The rapid increase in its large-$V$ magnitude with increasing $F$ confirms the enhancement in the EPC with the incorporation of AgNPs.

Figure 3 a shows the magnitude of $\lambda$, obtained from Eq. (1) using $d_{pc}dR_{pc}/dV$ values at large $V$, as function of $F$. Remarkably, we find $\lambda$ can be as high as $\approx 20$ for $F \sim 0.5 - 0.7$, before dropping to ~1, expected for a film of small AgNPs[29]. Both the magnitude and the $F$-dependence of $\lambda$ from point contact spectroscopy are consistent within 20% with the estimates obtained by fitting the $\rho - T$ data with the Bloch-Grüneisen formula at low $T$ ($\leq 100$ K) (open symbols in Fig. 3a, see Methods and Fig. S10 in Supplementary Information for details). Such large $\lambda$ is unprecedented in metallic solids and exceeds those with strong EPC, for example, the A15 compounds, by at least a factor of ~ ten (Fig. 3b, also see Supplementary Information section VIII for details). Intriguingly, the extended correlation between $\lambda$ and the normalized resistivity in Fig. 3b suggests Ag@Au hybrids to be an 'extreme' case of a metal where the electron-phonon scattering drives $\rho \to \rho_{MIR}$ even at room temperature.

Surface scattering can increase EPC in nanostructured metal films compared to bulk[22], but such enhancements are within a factor of ~ two, and thus much smaller than the enhancement in $\lambda$ observed here. The formation of the solution-processed Ag@Au core-shell nanoparticle hybrids and their stability against galvanic replacement is critically dependent on the interfacial charge transfer that results in the formation of electric dipoles across the hetero-interface[34–37]. The charge transfer can be parametrized from the difference in on-site energies ($\epsilon_0$) of Au and Ag, which is distributed as on-site excess charge $\delta n$ (bottom panel, Fig. 3c), and illustrated for a $4 \times 4$ Ag atom array embedded in an array of Au atoms (details in Supplementary Information Section IX). We experimentally verified such transfer of charge in our nanohybrids using X-ray photoelectron spectroscopy (XPS), where a red-shift in the Au ($4f_{5/2}$) with respect to neutral $Au^0$ ($4f_{5/2}$) suggests negative charge doping in Au (Fig. 3d). More detailed XPS results are available in Fig. S5 in Supplementary Information, which indicates average electron doping of the Au atoms by as much as

-0.6 ± 0.1 per atom for $F = 0.5$ (Supplementary Information Section II.B). The radial dipoles across the interface, formed when the $Ag^{|\delta n|+}$ and $Au^{|\delta n|-}$ sites assume opposite oxidation states can couple strongly to the lattice phonons via long-range Coulomb interactions. A detailed analytical and computational model on the two-dimensional array of Ag@Au hybrid indeed confirms additional contributions to the electron-phonon matrix elements, $g$ through the inter-site Coulomb interaction, thereby enhancing the Migdal-Eliashberg function $\alpha^2 \mathcal{F}(\omega)$ [Eq. (12)], and thus $\lambda$[38] (see inset of Fig. 3d, Methods and Supplementary Information Section IX).

We now focus on the resistivity saturation at $T \gg \Theta_D$, which probably is the 'smoking gun' signature of the strong emergent EPC in Ag@Au hybrids. In fact, our ability to vary $\lambda$ by over a factor ~200 (from bare gold film to Ag@Au nanohybrid at $F \approx 0.7$), allows access to the phonon contribution to resistivity dynamically from weak to ultra-strong coupling regime on a single material platform for the first time. In Fig. 4, we plotted the high-temperature segment (300 K $\geq T \geq$ 150 K, i.e., $T \geq \Theta_D$) of ($\rho - \rho_0$) shown in Fig. 1h for all $F$, where the temperature axis is scaled by the corresponding $\lambda$, obtained from the point contact measurements. Two key observations can be summarized as follows: First, the collapse of the resistivity traces for different $F$ onto a single one suggests $\lambda T$ would continue to be the 'scaling variable' that determines the resistivity even at very large $\lambda$, although the perturbative limit with linear scattering rate $\approx 2\pi k_B \lambda T/\hbar$, is expectedly recovered only when $\lambda \to 0$ (dashed line). Second, the sublinearity in ($\rho - \rho_0$) at large $\lambda T$, representing 'resistivity saturation', can be modelled with

$$\frac{1}{\rho - \rho_0} = \frac{1}{\rho_{BG}} + \frac{1}{\rho_\parallel}$$ (2)

which resembles a "parallel resistor channel" with $\rho_{BG}(T \geq \Theta_D) = 2\pi m k_B \lambda T/(\hbar n e^2)$[16,39] (See Methods and Supplementary Information V for details), and $\rho_\parallel$ is the resistivity of a parallel non-classical channel whose universality, temperature dependence, or even existence, have been questioned many times, but without a satisfactory answer so far[2,4,8]. The solid line fit in Fig. 4 corresponds to $\rho_\parallel \approx 20\,\mu\Omega \cdot m$ implying that the phonon contribution to resistivity can be described by an

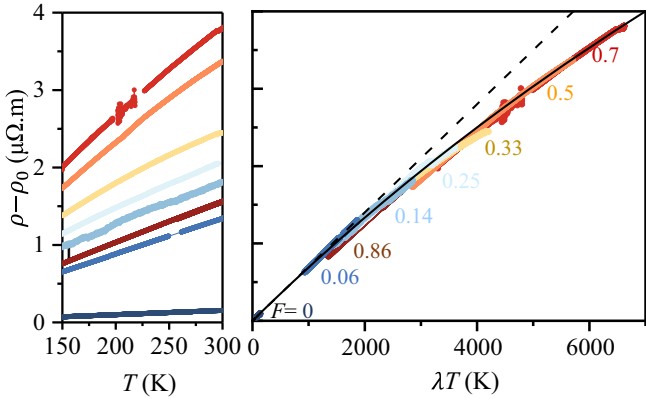

**Fig. 4 | Scaling of resisitivity and electron-phonon coupling strength.** Universality in the $T$-dependent component of $\rho$ for $T > \Theta_D$-150 K (left panel) for all $F$ with $T$ scaled with the corresponding EPC parameter $\lambda$ estimated from point contact spectroscopy measurements (right panel). The dashed line in the right panel indicates the 'Planckian' resistivity (=$2m\pi k_B \lambda T/(\hbar n e^2)$). The solid line is a fit to $(\rho - \rho_0)$ according to the two-component parallel channel model given by Eq. [(2)] (see text).

'ideal' Bloch-Grüneisen behaviour in parallel to a $T$-independent non-classical channel of resistivity close to the MIR-limit. In fact, the variation in $\rho(T)$ over the entire experimental temperature range can be satisfactorily captured by using the full form of $\rho_{BG}(T)$ and Eq. (2) (dashed lines in Fig. 1h). See Methods and Supplementary Information Section V for further discussions on the parallel resistor formula and other fit protocols.

Our experiment confirms the long-suspected inevitability of resistivity saturation in metals[3], irrespective of the strength of the EPC. However, there are deeper consequences. The first concerns the question, is there a universal bound to the EPC for a metal to exist? The persistence of metallic transport in Ag@Au hybrids with $\lambda \gg 1$, is *prima facie* at odds with a stability bound observed in Monte Carlo calculations on the Holstein model[15], or other fundamental 'Planckian limits' to dissipation or thermalization in metals[12,13]. An important consideration would, however, be the heterogeneous nature of our system, where the electrons couple to engineered vibration modes of a foreign species, such as the surface phonon modes of AgNPs. In this aspect, our system is fundamentally different from homogenous crystalline metals, which are unstable towards the formation of polaronic insulators at strong EPC. Nonetheless, the $T$-dependence of $\rho$ can also be fitted with a thermally activated parallel conduction channel (See Section V in Supplementary Information), and thus a possible coexistence of polaronic insulating phase and itinerant electrons cannot be ruled out[40-42]. Second, there is also a discrepancy with the models of resistivity saturation built on the breakdown of Born-Oppenheimer approximation and Matthiessen's rule, for example, polaronic deformation of disorder[43] or phonon-assisted delocalization[8,9,11]. These mechanisms often involve crossover to negative temperature coefficient of resistivity at $\rho \sim \rho_{MIR}$, i.e., 'Mooij correlation'[17,43], which was not observed in any of our samples. The absence of Anderson localization itself, even for the very strong disorder at $F \sim 0.5 - 0.8$ at low $T$ (down to ~0.3 K, See Fig. S7 in Supplementary Information), indicates that, unlike a conventional disordered metallic phase, the phase-coherent effects here are also modified. Finally, we also consider the role of long-range electron-electron interaction, which can impact transport in multiple ways, including assisting in the delocalization of carriers[44], introducing hydrodynamic viscosity[45], or even the suppression of resistivity saturation itself[2,4,5]. Such interaction is expected to be weak at metallic densities (measured by the low-field Hall effect (Fig. S9 in Supplementary Information)), although a modification of this scenario due to the presence of

many-body effects driven by phonons at the buried interfaces cannot be ruled out[46].

In conclusion, we have reported the realization of a metallic hybrid composed of ultra-small silver cores dispersed in a crystalline gold matrix, in which the electrical resistivity shows saturation as the silver core density and temperature are increased. Both electrical transport and point contact spectroscopy reveal that the electron-phonon coupling $\lambda$ in these engineered metallic hybrids can be as large as ~20, more than ten times than any known metallic solid. Our experiments outline a novel strategy to modify some of the fundamental properties of solids utilizing buried interfaces at the nanoscale.

## Methods

### Chemical synthesis

Ag@Au nanohybrids (Ag@Au NHs) were synthesized using a colloidal approach[31,47]. The synthesis process involved two sequential stages: the reduction of $AgNO_3$ with ice-cold $NaBH_4$ to form Ag nanoparticles (AgNPs) in an aqueous solution (in ultrapure water, purity-18.2 MΩ.cm) containing NaOH, $NH_4Br$, KI, and CTAB as a capping agent, followed by the introduction of $HAuCl_4$ at 40°C with continuous stirring. In situ, UV-Vis absorption spectroscopy was employed to monitor the synthesis process, and the reaction was terminated by adding isopropyl alcohol (IPA), resulting in nanohybrid agglomeration. The solution was then centrifuged at 10, 000 rpm for 15 minutes to remove excess CTAB and isolate the nanohybrids.

### Characterization

**UV-Vis Spectroscopy.** UV-vis absorption spectroscopy was used to assess the formation of AgNPs and Ag@Ag nanohybrids. The addition of $NaBH_4$ in the reaction mixture containing $AgNO_3$, at a specific 'wait time' ($t$) resulted in a prominent peak around ~393 nm, corresponding to the localized surface plasmon resonance (LSPR) absorption band of ultra-small AgNPs. Upon introducing $HAuCl_4$, the LSPR peak underwent a redshift to ~524 nm over approximately 1000 seconds, indicating the formation of a thicker shell due to spontaneous interdiffusion. To ensure a well-defined AgNP surface, we employed a strategic approach. Monitoring the SPR band shift with UV-Vis spectroscopy, we terminated the reaction by adding isopropyl alcohol (IPA) approximately 30 seconds after $HAuCl_4$ addition.

**X-Ray Photo-electron Spectroscopy.** An Axis Ultra K$\alpha$ X-ray photo-electron spectrometer with a monochromatized photon energy of ~1486.6 eV was used for all XPS measurements. To minimize moisture absorption, the sample was quickly inserted into the load-lock of the instrument, pumped in the entry chamber until the pressure around $10^{-8}$ mbar was reached, and subsequently transferred to the analysis chamber. The individual core-level spectra were corrected for charging using C-1$s$ peak at 284.5 eV as standard. The peak fitting of the individual core-levels was done using Casa XPS software.

**Transmission Electron Microscopy.** Transmission Electron Microscopy (TEM) was employed to examine the structural characteristics of an Ag@Au nanohybrid. The FEI TITAN Themis TEM operating at 300 kV, which offers a point resolution of ~0.2 nm and an energy spread of 0.136 nm, was utilized to capture all TEM images. High-Resolution Transmission Electron Microscopy (HRTEM) images were obtained along the zone axis to assess the Ag-Au interface and investigate any defects that may have arisen within the bimetallic entities. To prepare the sample for TEM imaging, the nanoparticle underwent multiple cleaning cycles in a chloroform ($CHCl_3$) : methanol mixture (1: 3 ratio), followed by centrifugation at 15000 rpm for 5 minutes. The resulting precipitated sample was then dispersed in chloroform and deposited onto a carbon-coated TEM grid, which was subsequently dried under vacuum overnight.

## Film preparation

The drop-cast technique was employed to fabricate the Ag@Au NH film onto prepatterned Cr/Au contacts (with thickness of $\approx 10$ nm/60 nm) arranged in various lead configurations on a glass substrate. Prior to drop-casting onto the pre-patterned leads, the sample was dissolved in CHCl$_3$. Subsequently, the sample was dried at 70 °C for 30 seconds and washed with deionized (DI) water followed by KOH solution and IPA to eliminate any excess CTAB and achieve a chemically sintered cross-linked nanostructure. This process was repeated ten times for each film, resulting in an average film thickness of $t_f \approx 3 \pm 0.5\,\mu m$ and a diameter of $\approx 4$ mm, typically covering the leads (Fig. 1e of Main Manuscript, Fig. S6 in Supplementary Information).

## Electrical Measurement

Four-probe resistivity of the sample was measured down to temperature ($T$) ~6 K in a home-built cryostat by passing a DC current of ~100 $\mu$A with Keithley 6221 and measuring the voltage with Keithley 2182A. The voltages across multiple contacts were recorded using the Keithley 3700 Multiplexer card. The voltage was measured in delta mode to cancel any thermo-emf across the contacts. Resistivity from $T$ ~10 K down to $T$~0.3 K was measured in a He3 cryostat.

## Fitting of $\rho - T$ data

The resistivity ($\rho$) of metal with electron-phonon interaction playing the dominant role of scattering can be expressed in terms of the Bloch Grüneisen[16,39] form as:

$$\rho(T) = \rho_0 + \rho_{BG}(T) \tag{3}$$

where $\rho_0$ is the residual resistivity, and

$$\rho_{BG} = \frac{2\pi\lambda k_B/\Theta_D}{(n/m)e^2}\left(\frac{T}{\Theta_D}\right)^5 \int_0^{\Theta_D/T} \frac{x^5}{(e^x-1)(1-e^{-x})}dx \tag{4}$$

is the Bloch Grüneisen form of resistivity arising from electron-phonon scattering. $\Theta_D$, the Debye temperature, and $\lambda$, the electron-phonon coupling constant can be estimated by fitting the $\rho - T$ data with Eq. [(3)]. As shown in Fig. 1h and Fig. S10 in Supplementary Information, we have fitted the $\rho - T$ data of AuNP and AgNP films using Eq. [(3)]. $\Theta_D$ ~ 170 K, and $\lambda$ ~ 0.45 for Au, and $\Theta_D$ ~ 190 K and $\lambda$ ~ 1 for Ag are estimated as fit parameters. The increased value of $\lambda$ as compared to the bulk value of ~0.2 for Au and Ag could be attributed to the nanostructuring in the film and increased electron scattering from the surfaces[22]. For Ag@Au films, Eq. [(3)] cannot describe $\rho - T$ for the entire range of $T$. Fig. S10 in Supplementary Information shows that the transport data for $F = 0.5$ deviates from the low-temperature BG fit ($T \leq 100$ K) to the data. For fitting the $\rho - T$ data for Ag@Au hybrid films, Eq. [2] is used where $\rho_{\parallel}$, $\Theta_D$ and $\lambda$ are the parameters of fit. $\Theta_D$ obtained from fitting $\rho - T$ of Ag@Au films with Eq. [(3)] in the low $T$ range, and with Eq. [2] with a parallel conduction channel, are consistent and lies within the range of $150 - 170$ K for all films (Fig. S10 in Supplementary Information). $\Theta_D$ being close to the Debye temperature of Au ($\Theta_{D,Au}$ ~ 170 K) in all cases indicates the electrical conduction occurs primarily within the host lattice of Au. $\lambda$ derived from the low-$T$ Bloch Grüneisen fit is shown in Fig. 3a. However, $\lambda$ estimated from parallel channel fit is slightly over-estimated (~20%) and probably less accurate since the scattering mechanism with this model, even at low $T$, is not purely electron-phonon mediated. $\rho_{\parallel}$ estimated from the parallel channel fit, given by Eq. [2], is plotted in Fig. S10d of Supplementary Information as a function of $F$ after normalizing with $\rho_{MIR}$. $\rho_{\parallel} \approx 8 - 25\,\mu\Omega$. m is $T$-independent, and lies within a factor of two of $\rho_{MIR}$ ~ 10 $\mu\Omega$.m, which drives the saturation of $\rho(T)$.

## Point contact measurements

**Experimental setup.** A sharp Pt/Rh metallic tip is brought in contact with the film in a controlled manner with the help of nanopositioners (attocubes and piezo tubes) as indicated in the schematic of the experimental set-up in Fig. 2(a) of main manuscript and Fig. S12 in Supplementary Information. The tip-sample chamber is loaded inside a home-built cryostat that could be cooled down to $T$ ~ 5 K.

**Theoretical framework for analysis.** Point contact measurements are done with the technique of modulation spectroscopy[33,48] which measures the higher-order derivatives of a signal through its AC components at harmonics of a definite frequency. The circuit for measurement of the point contact voltage is shown in Fig. S12 of Supplementary Information. A mixed AC+DC current $I + i_m \sin(\omega t)$ is passed through the sample using a constant current circuit. AC voltage from a lock-in amplifier and DC voltage from Keithley 2400 are added with an op-amp adder. A series resistance $R_s \gg R_{pc}$ is used to achieve the constant current in the circuit. The voltage across the tip-sample contact can be represented as a Taylor series expansion of the $I - V$ curve as follows:

$$
\begin{aligned}
V = f(I + i_m \sin(\omega t)) &= \sum_{k=1}^{\infty} \frac{i_m^k}{k!}\frac{dV^k}{d^kI}\sin^k(\omega t) \\
&= V + i_m \sin(\omega t)\frac{dV}{dI} + \frac{1}{2!}\frac{d^2V}{dI^2}i_m^2\sin^2(\omega t) \\
&\quad + \frac{1}{3!}\frac{d^3V}{dI^3}i_m^3\sin^3(\omega t) + ... \\
&= 1\left(V + \frac{i_m^2}{4}\frac{d^2V}{dI^2} + ...\right) \\
&\quad + \sin(\omega t)\left(i_m\frac{dV}{dI} + \frac{i_m^3}{8}\frac{d^3V}{dI^3} + ...\right) \\
&\quad - \cos(2\omega t)\left(\frac{i_m^2}{4}\frac{d^2V}{dI^2} + \frac{i_m^4}{12}\frac{d^4V}{dI^4} + ...\right) + ..
\end{aligned}
\tag{5}
$$

Grouping terms of the same amplitude of the modulation frequency $\omega$ gives

$$V = V_0 + \sum_{i=n}^{\infty}(a_{2n-1}\sin((2n-1)\omega t) + a_{2n}\cos(2n\omega t)) \tag{6}$$

For a small AC current $i_m \ll I$, the higher-order terms in the series expansion, which vary as $i^n$ can be neglected in each group and the amplitude of the voltage at frequency $n\omega$ becomes proportional to the $n$-th order derivative .i.e. $a_n \propto d^nV/dI^n$. Also, it is to be noted that the odd harmonics are in-phase (sine component) with the source AC signal $i_m \sin(\omega t)$, and even harmonics are out of phase, at 90° (cosine component) with the signal.

Hence, the first order derivative can be estimated from the amplitude of the $\omega$ component at 0° phase as in Eq. [(5)].

$$
\begin{aligned}
a_1 &= i_m\frac{dV}{dI} \\
\frac{dV}{dI} &= \frac{a_1}{i_m}
\end{aligned}
\tag{7}
$$

The second order derivative can be derived from the amplitude of the $2\omega$ component at 90° phase as in Eq. [(5)].

$$
\begin{aligned}
a_2 &= -\frac{i_m^2}{4}\frac{d^2V}{dI^2} \\
\frac{d^2V}{dI^2} &= -\frac{4a_2}{i_m^2}
\end{aligned}
\tag{8}
$$

Both $\omega$ and $2\omega$ components of the voltage difference across the tip-sample junction are acquired simultaneously with two lock-in amplifiers as denoted by $V_\omega$, and $V_{2\omega}$ in Fig. S12 of Supplementary Information respectively, after amplification with SR 560. The sample is grounded through current-voltage amplifier SR 570, which allows us to constantly monitor the current through the sample and hence tune the point contact resistance ($R_{pc}$).

Since a lock-in amplifier shows the rms value of a signal, the amplitude of the signal is $\sqrt{2}$ times the measured value i.e. $a_n = V_{n\omega}\sqrt{2}$, where $V_{n\omega}$ is the measured signal at $n\omega$. If $V_{AC}$ is the voltage at the sine output of the lockin amplifier, the AC current through the sample is $i_m = V_{AC}\sqrt{2}$. $V_\omega$, being the measured voltage by the lock-in amplifier at $\omega$ component and 0° phase, the amplitude of the voltage drop across the tip and sample is expressed using Eq. [(7)] as

$$\frac{dV}{dI} = \frac{a_1}{i_m} = \frac{V_\omega \sqrt{2}}{V_{AC}\sqrt{2}/R_s} = \frac{V_\omega}{V_{AC}/R_s} \quad (9)$$

Similarly, $V_{2\omega}$, being the signal measured by the lockin amplifier at $2\omega$ component and 90° phase, the second order derivative is expressed using Eq. [(8)] as

$$\frac{d^2V}{dI^2} = -\frac{4V_{2\omega}\sqrt{2}}{(\sqrt{2}V_{AC}/R_s)^2} = -\frac{4V_{2\omega}}{\sqrt{2}(V_{AC}/R_s)^2} \quad (10)$$

$dV/dI$ corresponds to the point contact resistance $R_{pc}$ and $d^2V/dI^2$ is related to the derivative of $R_{pc}$ with bias as $d^2V/dI^2 = R_{pc}dR_{pc}/dV$.

With appropriate multi-stage vibration isolation, $R_{pc}$ ranging from 500 $\Omega$ to 2 K$\Omega$ could be stabilized typically in the Ag@Au hybrid films with attocube and piezo controllers by monitoring the current through the sample. The modulation AC current $i_m$ was typically fixed at 1–5 $\mu$A, whereas the DC current/was varied till -100 −200 $\mu$A in magnitude. $R_{pc}$ can be expressed as a combination of ballistic Sharvin resistance ($R_{sh} = 16\rho l/3\pi d_{pc}^2$) and diffusive Maxwell resistance[33] ($R_M = \rho/d_{pc}$).

$$R_{pc} = \frac{16\rho l}{3\pi d_{pc}^2} + \frac{\rho}{d_{pc}} \quad (11)$$

where $d_{pc}$ is the diameter of the point contact orifice as shown in the schematic of Fig. 2b of the main manuscript. For an ideal ballistic point contact, the derivative of the point contact resistance represents the Migdal Eliashberg spectral function (Eq. [(12)]) $g(\epsilon) = \alpha^2\mathcal{F}(\epsilon)$[33], given by

$$\alpha^2\mathcal{F}(\omega) = \mathcal{G}(\omega) = \propto \sum_{k,q}|g(k,q)|^2\delta(\varepsilon_k)\delta(\varepsilon_{k+q})\delta(\omega - \omega_q) \quad (12)$$

where $k$ is the electron wave vector, $q$ and $\omega_q$ are the phonon wave vector and frequency, respectively; $A$ is a normalization constant and $g(k, q)$ are the electron-phonon matrix elements. The Migdal Eliashberg function represents the probability of specific phonon modes (with energy $\epsilon$) to decay into an electron-hole pair and closely resemble the phonon density of states in most cases.

$$\frac{1}{R_{sh}}\frac{dR_{pc}}{dV} = \frac{8ed_{pc}}{3hv_F}\mathcal{G}(\epsilon)|_{\epsilon = eV} \quad (13)$$

The integral of $\mathcal{G}(\epsilon)$ is a measure of the electron-phonon coupling constant.

$$\lambda = 2\int_0^\infty \frac{\mathcal{G}(\epsilon)}{\epsilon}d\epsilon \quad (14)$$

Hence, from the measurement of the point contact spectrum, we can estimate the electron-phonon coupling parameter.

An ideal ballistic contact is not achieved in experiments, and hence, one needs to consider non-equilibrium processes to analyze the point contact spectrum. However, $\lambda$ can be estimated from the non-zero background of the point-contact spectrum ($eV \gg \hbar\omega_D$), irrespective of the nature of the contact as long as phonons are the primary source of inelastic scattering, and elastic scattering dominates over the inelastic scattering processes. These conditions are satisfied in our system. See Supplementary Information Section VII for detailed derivations.

It can be shown $dR_{PC}/dV$ multiplied by the point contact diameter, $d_{PC}$ is directly proportional to $\lambda$, with the proportionality factor determined by the mass $m$, and number density $n$ of the electrons. Specifically, we get,

$$d_{PC}\frac{dR_{PC}}{dV}\bigg|_{V\to\infty} = \frac{16f}{3\pi}\frac{m}{neh}\lambda \quad (15)$$

where the constant $f \sim 1.1$–$1.8$, for a contact varying from diffusive to thermal regimes of transport. Using a typical value of resistivity for Ag@Au nanohybrid films $\rho \sim 10\,\mu\Omega$.m, we can estimate the mean free path, $l$ from the Drude expression $\rho = mv_F/ne^2l$ ($v_F$ is the Fermi velocity) as $l \approx 0.1$ nm, which is smaller than mechanically achieved point contacts. This suggests a diffusive/thermal nature of the contact since $l \ll d_{pc}$. $d_{PC}$ is estimated by assuming the point-contact resistance to be arising primarily from the Maxwell contribution .i.e $R_{PC} = \rho/d_{PC}$. The estimation of $d_{PC}$ is validated by the scaling of $\rho/d_{PC}\,dR_{PC}/dV$ at different values of $R_{PC}$, as shown in Fig. 2c of the main manuscript. Since for the typical values of point contact that we could achieve, $d_{PC} \lesssim l_{in}$, we have used $f \sim 1.1$, which is the case for the diffusive regime of point contact transport. Eq. [(15)] can be inverted to derive $\lambda$ as:

$$\lambda = \frac{3\pi}{16}\frac{neh}{m}\left[\frac{\rho}{R_{pc}}\frac{dR_{pc}}{dV}\right]_{V\to\infty} \quad (16)$$

Further details regarding the theoretical framework and analysis are discussed in Supplementary Information Section. VII

## Computational details

The theoretical calculations for the charge transfer were done on a 2D 'toy model' where a periodic array of (4 × 4) clusters of "Ag" sites are surrounded by 48 "Au" sites forming a superlattice of (8 × 8) 2D supercells. The calculations were carried out using the model Hamiltonian described in Supplementary Information Section IX. The relative ratio of the number of atoms for Ag:Au has been taken to be 1:3 to resemble a typical fraction $F = 0.25$ of the Ag@Au core@shell nanohybrid. Due to the mismatch of the local potential seen by the conduction electrons localized in Wannier orbitals at the Ag and Au sites, as shown in the top panel of Fig. 3c, and the long-range Coulomb interactions between electrons occupying these Wannier orbitals, there is a charge transfer at each site, indicated by the excess electron occupancy $\delta n$. The amount of charge transferred can be tuned to mimic the experimental values by varying the onsite potential difference between Au and Ag atoms, $\epsilon_0$.

As shown in Fig. 3c of the text, Au and Ag atoms become electron and hole-doped, respectively, for most positive values of $\epsilon_0$, which is also consistent with the X-ray photoelectron spectroscopy (XPS) data.

The parameters used in our study are as follows: the nearest neighbour distance, $d_1$, within the square lattice considered is set to 4.10 Å, corresponding to the lattice spacing in FCC-Ag/Au. The nearest neighbour hopping between all the atomic sites is set at 1 eV, and the hopping decay factor, $\xi_0 = d_1$, is adjusted to yield the second-nearest neighbour hopping of 0.5 eV. The on-site Coulomb interaction energy is fixed at $U_0 = 2$ eV, and $V_0$, which controls the strength of the long-range Coulomb interactions, is chosen such that $V_0/d_1 = 2$ eV. The

difference in work function between Au and Ag atoms is incorporated into the on-site potentials, with $\epsilon_0 = (\epsilon_j^{Ag} - \epsilon_j^{Au})$. The charge density obtained from solving the self-consistent mean field equations at $(\epsilon_j^{Ag} - \epsilon_j^{Au}) = 1$ eV is presented in the bottom panel of Fig. 3c of the main manuscript. More detailed figures are provided in the Supplementary Information Section IX.

Phonons are computed with spring constants of 1.0 for nearest neighbour atoms and 0.5 for next-nearest neighbours in arbitrary units. Electronic and phononic band spectra are calculated on $(16 \times 16)$ **k, q** grids within the Brillouin zone of the supercell and are used to determine the electron-phonon coupling strength $(\lambda^{(calc)})$ using the 'double delta' approximation. Variation of $\lambda^{(calc)}$ with $\langle \delta n_{Au} \rangle$, average electron occupancy on Au sites is shown in the inset of Fig. 3d, where $\langle \delta n_{Au} \rangle$ has been tuned by changing $\epsilon_0$ from 0.1 eV to 4 eV.

## Data availability
Data that support the plots within this paper, and other findings of this study are available from the corresponding author upon reasonable request. Source data are provided with this paper.

## Code availability
The codes that support the findings of this study are available from the corresponding author upon reasonable request.

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

## Acknowledgements

S.K. acknowledges support from the Prime Ministers' Research Fellowship (PMRF), Department of Science and Technology (DST), India. S.K. and B.D. gratefully acknowledge the Indian Institute of Science (IISC), Bangalore for funding. T.K.M. acknowledges the Science and Engineering Research Board (SERB)-National Postdoctoral Fellowship (NPDF), India for funding under Grant No. PDF/2021/001175. B.T. acknowledges DST Inspire Faculty, Grant No. SP/DSTO-21-0141 for funding. S.M., M.J., and H.R.K. thank the Supercomputer Education and Research Centre (SERC) at IISC, Bangalore, for providing the computational resources. M.J. acknowledges the National Supercomputing Mission (NSM) of DST, India, and Nano Mission (NM) Council of DST, India,for financial support under Grants No. DST/NSM/R&D_HPC_Applications/2021/23 and No. DST/NM/TUE/QM-10/2019 respectively. H.R.K. gratefully acknowledges support from the Indian National Science Academy (INSA) under Grant No. INSA/SP/SS/2023/, the SERB-DST, India, under Grant No. SB/DF/005/2017/, and at the International Centre for Theoretical Sciences (ICTS) from the Simons Foundation (Grant No. 677895, R.G.). A.P. and A.G. acknowledge the IISC-Institute of Eminence (IOE) for funding and support and Prof. Anurag Kumar.

## Author contributions

S.K. and T.K.M. contributed equally to this work. S.K. performed the electrical transport and point contact spectroscopy measurements with help from T.P.S. T.K.M fabricated the devices and performed material characterization with help from S.K. and B.D., B.T. performed the structural characterization of the samples using transmission electron microscopy with the help of T.K.M. S.M., H.S., M.J., and H.R.K. provided the presented theory calculations. A.P. provided experimental inputs to the sample fabrication. A.G. contributed to the data interpretation and theoretical understanding of the manuscript. S.K., T.K.M., and A.G. wrote the manuscript with inputs from all authors.

## Competing interests

The authors declare no competing interests.
