## [Transparent Peer Review file · Nature Communications]

Engineering ultra-strong electron-phonon coupling and nonclassical electron transport in crystalline gold with nanoscale interfaces

Corresponding Author: Ms SHREYA KUMBHAKAR

Version 0:

Reviewer comments:

Reviewer #1

(Remarks to the Author)

The manuscript "Engineering ultra-strong electron-phonon coupling and nonclassical electron transport in crystalline gold with nanoscale interfaces " by Kumbhakar et al. reports on the first realization of the metallic system where the electron-phonon interaction can be effectively tuned and increased by an incredible factor of 100. It is due to embedding a network of nm sized Ag particles of varying density inside a crystalline Au shell which presumably ensures a strong coupling of conduction electrons to the localized phonon modes. If so, many important physical questions of transport properties in metals can be addressed in unprecedented limits of the parameter of the electron-phonon interaction. In this context the paper brings a very important and timely subject.

Experimentally, upon increase of the Ag nanoparticles density, the resistivity of the samples increases by two orders but also temperature dependence of the resistivity changes from linear at low density to sublinear behavior at higher density approaching a saturation to the Mott-Ioffe-Regel scale. The analysis of the resistivity data leads the authors to the conclusion that the electron-phonon coupling λ in these engineered metallic hybrids can be as large as ~ 20 , more than ten times than in any known metallic solid. This is a major finding of the paper. It is supported by the point-contact spectroscopy (PCS) which is appropriate technique to study of the electron-phonon interaction in metals [Ref. 32 in the manuscript]. The authors properly analyze two spectroscopic PCS regimes which are indeed capable to provide an information on λ , but they completely omitted the thermal regime which is unfortunately most probably characteristic for their point contacts made on extremely high resistivity samples. Indeed, one has to distinguish elastic I_{el} and inelastic I_{in} electron's mean free path. In the diffusive (still spectroscopic) regime an elastic mean free path is much smaller than the diameter $\langle d \rangle$ of the PC orifice, but the distance at which the energy is released (square root of the product of I_{el} and I_{in}) is bigger than $\langle d \rangle$. In the thermal regime both I_{el} , $I_{in} \ll \langle d \rangle$ and the conduction electrons are not gaining an energy eV from applied voltage V because they lose its excess energy many times when passing through PC. Then, by increasing voltage one rather increase the local temperature and the measurements of $dV/dI(V)$ (Fig. 2(b)) is similar to the measurement of the temperature dependence of the differential resistance.

Before elucidating the point-contact regime it is not possible to make any conclusions on the electron-phonon coupling.

Reviewer #2

(Remarks to the Author)

In this revised version of the manuscript the authors have addressed my minor concerns from the previous round of review. I think the paper can be accepted to Nature Communications as-is.

My view of the objections raised by the other referees is as follows.

Reviewer #3 maintains that the "description of the methodology is still unclear," but my view of the current version of the manuscript is that it contains a good amount of detail about the techniques being used and their theoretical justification.

Reviewer #4 has a more substantive disagreement, which I summarize as follows. Referee #4 points out that (1) a modified T-dependence of $R(T)$ does not necessarily imply a strongly renormalized electron-phonon coupling, and (2) such a large electron-phonon coupling constant λ should (theoretically) lead to insulating behavior, and therefore not be possible in a metal.

My view of objection (1) is that it would be perfectly correct if the authors had only presented bulk transport measurements. But perhaps the primary virtue of this work is that they have presented point contact spectroscopy (PCS) results, which also show a huge value of λ . In this revision the authors have dedicated a significant amount of space to justifying why their PCS measurement gives a direct measure of λ . I cannot find any mistake in their reasoning (which I acknowledge is not the same thing as saying there is no mistake), so I have no grounds for disputing their interpretation of this measurement. Reviewer #4 did not mention the PCS measurements in their report, so I also do not see a reason from that report for not accepting the authors' interpretation.

Objection (2) is, in my opinion, one of the major positive reasons why this paper should be published. There is a definite theory expectation that such large values of λ should not be possible in a system with metallic conduction. The authors seem to have measured such a large value of λ . I think it would be unscientific to use this theory expectation as a reason to not publish the experiment (i.e., to force the experiment to conform to theory expectation). Perhaps there is some subtle reason (due to the nano-interfaces, perhaps) why this experiment evades the usual bounds set by the theory, or perhaps there is some subtle reason why the interpretation of the experiment is complicated, and what they call " λ " should not be thought of as equivalent to the usual electron-phonon coupling constant in metals. But either way I think the authors have found an interesting and provocative result and it should be published even if it "feels wrong" in the sense of violating a theory expectation.

Version 1:

Reviewer comments:

Reviewer #1

(Remarks to the Author)
Referee report

In the revised manuscript "Engineering ultra-strong electron-phonon coupling and nonclassical electron transport in crystalline gold with nanoscale interfaces" by Kumbhakar et al., the authors addressed my main concern about the point contact regime. It is the thermal one in their case and not the diffusive one as originally claimed. In the new version they properly show that while the spectroscopic information on the electron-phonon function is lost the general information on the electron-phonon coupling constant can be estimated also in the thermal regime. The detailed explanation is provided in the Supplementary information, Section V. With this I suggest to publish the paper in the Nature Communications.

There are errors/misprints in the Supplementary Information Section V., which most probably do not spoil the overall analysis but should be corrected:

- In
- A. Electron-phonon coupling ...
1. Ballistic regime
- Diameter of the PC is denoted once as d_{PC} but next as d
 - In Wexler formula (S8) Γ is equal to unity at $\lambda_{el} \ll d_{PC}$ which is the diffusive NOT THE BALLISTIC regime
 - Eq. S10 is reversed but then correctly used in Eq. S24.
 - Summary after 3. Thermal regime, first bullet wrongly reads Eq. 11a instead of Eq. S19a
- B. Analysis of experimental data
- Typical residual resistivities should be some 10^{-6} Ohm.m AND NOT 10^{-6} microOhm.m as stated
 - the estimated mean free path (0.1 nm) can hardly be larger than the typical PC diameter
 - λ_{e-ph} is defined as the Fermi velocity divided by the e-ph relaxation time?

Response to Reviewers' comments: NCOMMS-24-44901-T
Current Manuscript ID: NCOMMS-24-44901A

RESPONSE TO REVIEWER-1

The manuscript “Engineering ultra-strong electron-phonon coupling and nonclassical electron transport in crystalline gold with nanoscale interfaces “ by Kumbhakar et al. reports on the first realization of the metallic system where the electron-phonon interaction can be effectively tuned and increased by an incredible factor of 100. It is due to embedding a network of nm-sized Ag particles of varying density inside a crystalline Au shell, which presumably ensures a strong coupling of conduction electrons to the localized phonon modes. If so, many important physical questions of transport properties in metals can be addressed in unprecedented limits of the parameter of the electron-phonon interaction. In this context the paper brings a very important and timely subject. Experimentally, upon the increase of the Ag nanoparticles density, the resistivity of the samples increases by two orders, but also temperature dependence of the resistivity changes from linear at low density to sublinear behaviour at a higher density approaching a saturation to the Mott-Ioffe-Regel scale. The analysis of the resistivity data leads the authors to the conclusion that the electron-phonon coupling λ in these engineered metallic hybrids can be as large as ~ 20 , more than ten times than in any known metallic solid. This is a major finding of the paper. It is supported by point-contact spectroscopy (PCS), which is the appropriate technique to study the electron-phonon interaction in metals [Ref. 32 in the manuscript].

We thank the reviewer for pointing out the novelty of our work and the crucial role of point contact spectroscopy in providing information about the electron-phonon scattering of metals.

The authors properly analyze two spectroscopic PCS regimes which are indeed capable of providing information on λ , but they completely omitted the thermal regime, which is unfortunately most probably characteristic for their point contacts made on extremally high resistivity samples. Indeed, one has to distinguish elastic l_{el} and inelastic l electron’s mean free path. In the diffusive (still spectroscopic) regime, an elastic mean free path is much smaller than the diameter d of the PC orifice, but the distance at which the energy is released (square root of the product of l_{el} and l) is bigger than d . In the thermal regime both $l_{el}, l \ll d$ and the conduction electrons are not gaining an energy eV from applied voltage V because they loose its excess energy many times when passing through PC. Then, by increasing voltage, one rather increases the local temperature, and the measurements of $dV/dI(V)$ (Fig. 2(b)) are similar to the measurement of the temperature dependence of the differential resistance. Before elucidating the point-contact regime it is not possible to make any conclusions on the electron-phonon coupling.

The reviewer has rightly pointed out that we have not discussed the possibility of the thermal regime of point-contact transport in the results. While we do not completely agree that no information can be gained from the thermal regime, we have now elaborated our analysis and, thereby, our conclusions in the following ways.

1. We first present a detailed derivation of the point contact (PC) spectrum in all three regimes; ballistic, diffusive, and thermal and show that the electron-phonon coupling constant (λ) can be estimated from the background of the PC spectrum (at bias larger than the maximum phonon energy, which is close to the Debye energy), irrespective of the particular regime of transport. It is, however, the spectral information that is lost in the thermal regime, as has been correctly pointed out by the reviewer as well and gradually smeared out with an increase of non-equilibrium phonons even in the ballistic/diffusive regime. The derivations have been adapted from the references [1–4]. This analysis has now been included in the Supplementary Information Section. V.

The above results clearly show that the limit of strong reabsorption of phonons, also known as the generation of non-equilibrium phonons in the diffusive regime, gives an equivalent point contact spectrum as in the thermal regime when the elastic scattering significantly dominates the resistivity. Because of this we have so far labelled our point-contact as diffusive, which has been now changed to a **quasi-thermal point-contact regime** in the main manuscript for clarity.

2. We show additional data to elucidate the regime of point contact transport. From direct measurements of both the inelastic and elastic scattering lengths, and comparing those with the estimated diameter of the point contact, we find our experimental regime lies at the crossover from the diffusive to the thermal regimes. We call this the quasi-thermal regime and have modified the descriptions in the manuscript accordingly.

We emphasize that the net coupling constant, λ , remains essentially unchanged irrespective of the particular regime of the point contact, and all we lose is the spectral resolution, which is not under the current investigation anyway.

ELECTRON PHONON COUPLING FROM POINT CONTACT MEASUREMENTS

Point contact spectroscopy is the technique of extracting spectral information about the electron scattering mechanisms via transport through a narrow metallic constriction [1]. The narrowness of the constriction, quantified by its dimension $\sim d_{\text{PC}}$, as compared to the mean free path of the system, l_e (elastic mean free path), and l_{in} (inelastic mean free path) determines the efficiency of the energy-resolving spectroscopy. Below, we provide a detailed derivation of the point contact spectrum.

A. Ballistic regime

When the dimension of the point contact (PC), d_{PC} , is lesser than both the elastic and inelastic mean free paths *i.e.* $d_{\text{PC}} \ll l_{\text{el}}, l_{\text{in}}$, it is classified to be in the ballistic transport regime. Electrons passing through an ideal ballistic point-contact, biased with an external voltage V , gain excess energy eV due to the absence of inelastic collisions at the junction. This energy is then dissipated by scattering, creating quasiparticle excitations. The scattering events result in the reflection of the electrons to the PC junction, resulting in a backflow current. This leads to nonlinear $I - V$ characteristics, whose higher order derivatives can be shown to correspond to the interaction function causing the scattering. In this case, the bias, eV , gives the spectral resolution of the interaction function. For metals, the electron-phonon interaction (EPI) is generally the dominant source of scattering. EPI function, also known as the Migdal Eliashberg function $\alpha^2\tilde{\mathcal{F}}(\omega)$ can thus be estimated from the non-linearities of $I - V$ characteristics in a point contact.

The point contact resistance R_{PC} [1] of has been derived by Wexler to be a combination of the ballistic Sharvin resistance ($R_{\text{sh}} = 16\rho l/3\pi d_{\text{PC}}^2$) and diffusive Maxwell resistance ($R_{\text{M}} = \rho/d$) as follows

$$\begin{aligned} R_{\text{PC}} &= R_{\text{sh}} + \Gamma \left(\frac{l_{\text{el}}}{d} \right) R_{\text{M}} \\ &= \frac{16\rho l}{3\pi d^2} + \Gamma \left(\frac{l_{\text{el}}}{d} \right) \frac{\rho}{d} \end{aligned} \quad (1)$$

where Γ is a function of l_{el}/d . When $l_{\text{el}} \ll d_{\text{PC}}$, which is the ballistic regime $\Gamma \approx 1$. l in the above expression is the net mean free path of the system, which is often derived by Matthiessen's rule in noble metals $l^{-1} = l_{\text{el}}^{-1} + l_{\text{in}}^{-1}$. We can estimate $l = v_F\tau$ (τ being the electron scattering time scale) from the Drude resistivity, $\rho = m/ne^2\tau = mv_F/ne^2l$, which gives $R_{\text{sh}} = 16mv_F/3\pi ne^2d_{\text{PC}}^2$. Hence, Eq. [1] can be written as:

$$R_{\text{PC}} = R_{\text{sh}} \left(1 + \frac{3\pi d_{\text{PC}}}{16v_F\tau} \right) \quad (2)$$

It is useful to note the ratio of R_{sh} to R_{M} .

$$\frac{R_{\text{sh}}}{R_{\text{M}}} = \frac{3\pi d_{\text{PC}}}{16l} \quad (3)$$

Differentiating Eq. [2] gives

$$\frac{dR_{\text{PC}}}{dV} = \frac{3\pi d_{\text{PC}}}{16v_F} \frac{d}{dV} \left(\frac{1}{\tau} \right) R_{\text{sh}} \quad (4)$$

If the spectral dependence of the scattering comes entirely from the EPI, using Fermi-Golden rule we can write τ as

$$\begin{aligned} \frac{1}{\tau} &= \frac{2\pi}{\hbar} \int_0^{eV} \mathcal{G}(\epsilon) d\epsilon \\ \frac{d(1/\tau)}{dV} &= \frac{2\pi}{\hbar} e\mathcal{G}(\epsilon)|_{\epsilon=eV} \end{aligned} \quad (5)$$

where $\mathcal{G}(\epsilon)$ represents the Migdal-Eliashberg spectral function. Substituting this in Eq. [4] we have

$$\frac{1}{R_{\text{sh}}} \frac{dR_{\text{PC}}}{dV} \approx \frac{3ed_{\text{PC}}}{\hbar v_F} \mathcal{G}(\epsilon)|_{\epsilon=eV} \quad (6)$$

A more rigorous derivation by Kulik shows the PC spectrum to be

$$\frac{1}{R_0} \frac{dR_{\text{PC}}}{dV} = \frac{8}{3} \frac{ed_{\text{PC}}}{\hbar v_F} \mathcal{G}(\epsilon)|_{\epsilon=eV} \quad (7)$$

R_0 is the point contact resistance at zero bias and is equal to the Sharvin contribution in the ballistic regime, $R_0 = R_{\text{sh}}$. The electron-phonon coupling constant λ is

$$\lambda = 2 \int_0^\infty \frac{\mathcal{G}(\epsilon)}{\epsilon} d\epsilon \quad (8)$$

However, it has been observed in experiments that the PC spectrum often shows a non-zero background at higher voltages, which is of the order of $\mathcal{G}(\epsilon)$. This is not captured by Eq. [7] since $\mathcal{G}(\epsilon) = 0$ for $\epsilon \geq \hbar\omega_D$, ω_D being the Debye frequency. It has been shown by Gelder [2] for the first time and later demonstrated by Kulik [1, 3, 4] that if the point contact region is inhomogenous in geometry, inelastic collisions can happen at the narrowest part of the point contact. The relaxation of phonons spontaneously emitted in such collisions is a relatively slower process than the characteristic electron-phonon scattering time scales, leading to a phenomenon called *trapping/reabsorption of non-equilibrium phonons*. In other words, a non-equilibrium phonon gas at the PC orifice is generated due to the stimulated emission of phonons by the spontaneously emitted phonons. The electron scattering length scale by the non-equilibrium phonons, l_r is smaller than characteristic electron-phonon scattering length scales $l_{e-\text{ph}}$. The point contact spectrum [Eq. 7] in such cases is modified as [1]

$$\frac{1}{R_0} \frac{dR_{\text{PC}}}{dV} = \frac{8}{3} \frac{ed_{\text{PC}}}{\hbar v_F} [\mathcal{G}(\epsilon) + \gamma \int_0^\infty \frac{\mathcal{G}(\omega)}{\omega + \omega_0} d\omega + \frac{\gamma}{2} \frac{eV}{eV + \omega_0} \mathcal{G}(eV)] \quad (9)$$

where γ is a geometrical factor arising from the shape of the PC junction and is ≈ 0.58 for a PC orifice. $\omega_0 = \omega_D l_r l_{e-\text{ph}} / d_{\text{PC}}^2$ is the phonon reabsorption frequency. The last two terms represent correction to the ballistic expression, Eq. [7] due to a *background* signal.

B. Diffusive regime

A point contact is described as diffusive if the dimension of the PC, d_{PC} , is larger than the elastic mean free path (l_{el}) but smaller than the inelastic relaxation length during the diffusion motion of electrons in the contact, $\Lambda = \sqrt{l_e l_{\text{in}}/3}$, *i.e.* $l_{\text{el}} \ll d_{\text{PC}} \ll \sqrt{l_e l_{\text{in}}/3}$. This, of course, automatically implies $d < l_{\text{in}}$. Kulik has shown that even in the diffusive regime, spectral information about the electron-phonon interaction can be obtained, and the proportionality between dR_{PC}/dV and $\mathcal{G}(\epsilon)$ is preserved. However, the intensity of the PC spectrum deviates from Eq. [7] and is reduced by a factor K , known as the Knudsen factor, that depends on l_{el}/d . For $l_{\text{el}} \ll d$, it can be shown that $\langle K \rangle \sim (3\pi/4) l_{\text{el}}/d_{\text{PC}}$ in a PC orifice [1].

This modifies Eq. [7] as:

$$\frac{1}{R_0} \frac{dR_{\text{PC}}}{dV} = \frac{3\pi}{4} \frac{l_{\text{el}}}{d_{\text{PC}}} \frac{8ed_{\text{PC}}}{\hbar v_F} [\mathcal{G}(\epsilon) + \gamma \int_0^\infty \frac{g(\omega)}{\omega + \omega_0} d\omega + \frac{\gamma}{2} \frac{eV}{eV + \omega_0} \mathcal{G}(eV)] \quad (10)$$

For $l_e \ll d \ll l_{\text{in}}$, $R_0 \approx R_M \approx \rho/d$. Also, $l \approx l_{\text{el}}$. Hence, we can write Eq. [10] as follows:

$$\begin{aligned} \frac{1}{R_{\text{sh}}} \frac{dR_{\text{PC}}}{dV} &= \frac{R_M}{R_{\text{sh}}} \frac{3\pi l_{\text{el}}}{4d_{\text{PC}}} \frac{8ed_{\text{PC}}}{3\hbar v_F} [\mathcal{G}(\epsilon) + \gamma \int_0^\infty \frac{\mathcal{G}(\omega)}{\omega + \omega_0} d\omega + \frac{\gamma}{2} \frac{eV}{eV + \omega_0} \mathcal{G}(eV)] \\ &= \frac{3\pi d_{\text{PC}}}{16l_{\text{el}}} \frac{3\pi l_{\text{el}}}{4d_{\text{PC}}} \frac{8ed_{\text{PC}}}{3\hbar v_F} [\mathcal{G}(\epsilon) + \gamma \int_0^\infty \frac{g(\omega)}{\omega + \omega_0} d\omega + \frac{\gamma}{2} \frac{eV}{eV + \omega_0} \mathcal{G}(eV)] \\ &= \frac{3\pi^2}{8} \frac{ed_{\text{PC}}}{\hbar v_F} [\mathcal{G}(\epsilon) + \gamma \int_0^\infty \frac{\mathcal{G}(\omega)}{\omega + \omega_0} d\omega + \frac{\gamma}{2} \frac{eV}{eV + \omega_0} \mathcal{G}(eV)] \end{aligned} \quad (11)$$

Interestingly, this is equivalent to Eq. [9] derived before, representing that $1/R_{\text{sh}} dR_{\text{PC}}/dV$, not $1/R_0 dR_{\text{PC}}/dV$, is the correct measure of $\tilde{g}(\epsilon)$. From Eq. [9] and Eq. [11], we observe that in both the ballistic and diffusive regime, the point contact spectrum dR_{PC}/dV leads us to the EPI function.

However, for strong reabsorption of phonons, $\omega_0 \rightarrow 0$ since as $d \gg l_{e-\text{ph}}, l_r$. In such cases, the point contact spectrum

in both ballistic and diffusive regimes can be summarized in terms of the following expressions

$$\frac{1}{R_{\text{sh}}} \frac{dR_{\text{PC}}}{dV} \approx \frac{3\pi^2}{8} \frac{ed_{\text{PC}}}{\hbar v_F} \mathcal{G}(eV), \quad \epsilon \leq \hbar\omega_D, \text{ weak phonon reabsorption} \quad (12a)$$

$$\approx \frac{3\pi^2}{8} \frac{ed_{\text{PC}}}{\hbar v_F} [\mathcal{G}(eV)[1 + \frac{\gamma}{2}] + \frac{\gamma\lambda}{2}], \quad \epsilon \leq \hbar\omega_D, \text{ strong phonon reabsorption} \quad (12b)$$

$$\approx 0, \quad \epsilon \geq \hbar\omega_D, \text{ weak phonon reabsorption} \quad (12c)$$

$$\approx \frac{3\pi^2}{8} \frac{ed_{\text{PC}}}{\hbar v_F} \frac{\gamma}{2} \lambda, \quad \epsilon \geq \hbar\omega_D, \text{ strong phonon reabsorption} \quad (12d)$$

Since $\gamma \sim 0.58$ for a circular orifice, the background signal at large bias directly corresponds to the electron-phonon coupling constant as:

$$\boxed{\frac{1}{R_{\text{sh}}} \frac{dR_{\text{PC}}}{dV} (eV \gg \hbar\omega_D) \approx 1.1 \frac{ed_{\text{PC}}}{\hbar v_F} \lambda} \quad (13)$$

C. Thermal regime

The thermal regime of point contact is defined as $d_{\text{PC}} \ll l_{\text{el}}, l_{\text{in}}$. The electrons dissipate the energy at the constriction itself due to inelastic collisions, leading to the heating of the point contact. The temperature (T)-dependence of the resistivity ρ is the origin of non-linearities in $I - V$ in such cases and can be represented as [1]:

$$I(V) = Vd \int_0^1 \frac{dx}{\rho(T\sqrt{(1-x^2)})|_{T=eV/3.63k_B}} \quad (14)$$

This allows both calculation of the $I - V$ characteristic using $\rho(T)$ and vice-versa reconstruction of the $\rho(T)$ dependence in the constriction from the measured $I - V$ curve.

However, if the phonon contribution to the resistivity is less than the elastic scattering and the heat influx and heat outflux are both determined by electron-phonon collisions; then, a spatially dependent phonon distribution, related to $\mathcal{G}(\omega)$ will be introduced in the point-contact region, which is determined by the local temperature $T(r)$. This indicates that the argument of ρ in the above equation involves a spatial integration over $T(r)$. Hence, the second-order derivative of $I - V$, $d^2I/dV^2 \approx 1/R_0^2 dR_{\text{PC}}/dV$ can be shown to be related to $T(r)$, and equivalently $g(\omega)$. The PC spectrum in such cases was derived [5] as follows:

$$\frac{1}{R_0} \frac{dR_{\text{PC}}}{dV} = \frac{\pi\sqrt{3}m}{ne\hbar\rho} \int_0^\infty \frac{d\omega}{\omega} \mathcal{G}(\omega) S(eV/\omega) \quad (15)$$

where,

$$S(x) = \frac{2\pi}{3} \frac{d^2}{dx^2} \int_0^{\pi/2} \frac{dy}{\sinh^2(\pi/\sqrt{3}x\sin y)}$$

It is interesting to note the difference between the two extreme regimes: ballistic and thermal. In the former case, dR_{PC}/dV is proportional to $\mathcal{G}(\epsilon)$, whereas in the latter, it is proportional to $\mathcal{G}(\epsilon)$ convoluted by a function of $S(\epsilon)$, which arises from non-equilibrium or thermal effects. *Thus, the spectral information of the EPI is smeared out in the thermal regime.*

For $eV \gg \hbar\omega_D$, $S(eV/\hbar\omega) \approx 1$, at all finite values $\mathcal{G}(\omega)$. Hence, from Eq. [15] we can write:

$$\begin{aligned} \frac{1}{R_0} \frac{dR_{\text{PC}}}{dV} (eV \gg \hbar\omega_D) &= \frac{\pi\sqrt{3}m}{ne\hbar\rho} \int_0^{\hbar\omega_D} \frac{g(\omega)}{\omega} d\omega \\ &= \frac{\pi\sqrt{3}m}{2ne\hbar\rho} \lambda \end{aligned} \quad (16)$$

Since $R_{\text{PC}} \approx R_{\text{M}} = \rho/d$ in the thermal regime, the above equation can be written as

$$\frac{1}{R_{\text{sh}}} \frac{dR_{\text{PC}}}{dV} (eV \gg \hbar\omega_D) = \frac{R_{\text{M}}}{R_{\text{sh}}} \frac{\pi\sqrt{3}m}{2ne\hbar\rho} \lambda = \frac{3\pi d_{\text{PC}}}{16l} \frac{\pi\sqrt{3}m}{2ne\hbar\rho} \lambda = \frac{3\pi^2}{32} \frac{d_{\text{PC}}m}{ne\hbar} \frac{ne^2}{mv_F} \lambda \quad (17)$$

Figure 1. Variation of $S(x)$

So we arrive at the expression:

$$\frac{1}{R_{\text{sh}}} \frac{dR_{\text{PC}}}{dV}(eV \gg \hbar\omega_D) = 1.8 \frac{ed_{\text{PC}}}{\hbar v_F} \times \lambda \quad (18)$$

Summary

- In the ballistic and diffusive regimes, due to the absence of inelastic collisions at the point-contact the excess energy eV gained by the electrons while passing the PC provides the spectroscopic probe to the EPI, as represented by Eq. 11a.
- In the presence of inhomogeneities near the point contact, the spectrum deviates from the equilibrium case discussed above in the ballistic and diffusive regimes. The non-equilibrium processes change the effective PC temperature and result in a non-zero background as well as lead to smearing of the spectral dependence, as can be observed from Eq. [9], and Eq. [11]. In fact, for significant phonon trapping/reabsorption ($\omega_0 \rightarrow 0$), the background signal ($dR_{\text{PC}}/dV(eV \gg \hbar\omega_D)$) directly manifests the cumulative EPI, represented by λ (Eq. [13]).
- The ballistic and diffusive regimes are equilibrium conditions, where the non-equilibrium processes are considered as perturbations. Eq. [12(b),(d)] is the extreme limit of the perturbation. The thermal regime is considered to be intrinsically in the non-equilibrium regime when the point contact is smaller in size compared to both the elastic and inelastic collisions. In such a case, the spectral information is entirely smeared. However, the background value similarly corresponds to the net EPC value, Eq. [18].
- Eq. [13,18] can be summarized as

$$\frac{1}{R_{\text{sh}}} \frac{dR_{\text{PC}}}{dV}(eV \gg \hbar\omega_D) = f \frac{ed_{\text{PC}}}{\hbar v_F} \times \lambda \quad (19)$$

where the constant $f \sim 1.1 - 1.8$. **A remarkable implication of this expression is the fact that in the presence of a non-equilibrium phonon distribution at the point contact, the background signal in the PC spectrum represents the λ , irrespective of the origin of this non-equilibrium gas, an inhomogenous PC (ballistic/diffusive) or a ‘wide’ PC (thermal).** Physically, this means the scattering of electrons causing the non-linearities in $I - V$ at $eV \gg \hbar\omega_D$, is a cumulative contribution from the non-equilibrium phonons available at all energies below $\hbar\omega_D$. Hence, in all three regimes of PC transport, λ can be extracted from the background signal of the PC spectrum given phonons are the dominant inelastic scatterers in the system.

Eq. [19] can be rewritten as

$$d_{\text{PC}} \frac{dR_{\text{PC}}}{dV} \Big|_{V \rightarrow \infty} = \frac{16f}{3\pi} \frac{m}{ne\hbar} \lambda \quad (20)$$

The right side of the above equation is the PC spectrum normalized by the PC dimension and depends on physical entities intrinsic to the system. Eq. 20 can be inverted to derive the λ as shown in Eq. [1] in the main manuscript.

$$\lambda = \frac{3\pi}{16f} \frac{ne\hbar}{m} \left[d_{\text{PC}} \frac{dR_{\text{PC}}}{dV} \right]_{V \rightarrow \infty} \quad (21)$$

D. Analysis of experimental results

1. Estimation of the point contact diameter

The typical residual resistivities of the film that we have measured vary within $\rho \sim 10^{-6} - 10^{-5} \mu\Omega.m$. Assuming the fermi velocity of Au/Ag $\sim v_F \sim 1.4 \times 10^6 m.s^{-1}$, we get the mean free path to be $l = v_F\tau \sim 10^{-9} - 10^{-10} m$, where the electron scattering time, τ is estimated from the Drude expression of resistivity $\rho = m/ne^2\tau$. This is larger than the typical PC diameter achieved experimentally by mechanical methods. Hence, we expect the resistance to primarily dominate the Maxwell contribution *i.e.*, $R_{PC} \sim \rho/d_{PC}$. To establish this experimentally, we have performed the following experiment. The point contact resistance, R_{PC} was recorded in a film with Ag-filling fraction, $F = 0.7$ while approaching the tip towards the sample with a z-piezo positioner. R_{PC} kept on decreasing as the tip gradually approached the sample. Without loss of generality, the PC contact region can be modelled as a sphere of radius d_{PC} , with the centre of the sphere located at the centre of the PC-plane. This is depicted schematically in Fig. 2(a), where the side view shows a conical tip touching the sample. As the tip is approached further with the piezo, the sphere effectively becomes larger, thus decreasing R_{PC} . We have used a z-piezo from PI ceramics, whose axial displacement for a voltage increment of $0.1 V$ is $\sim 0.6 \text{ \AA}$. Fig. b shows the R_{PC} as a function of the net piezo displacement from initial conditions, which is equivalent to the change in PC dia, Δd_{PC} . The dotted line represents the Maxwell contribution, where d_0 is the starting PC diameter estimated as $d_0 \sim \rho/R_{\Delta d=0}$. The agreement of the dotted line with the experimental values represents the PC diameter to be accurately captured by the Maxwell resistance.

Figure 2. **Estimation of the point-contact diameter:** (a) Schematic of the experimental realization of the tip-sample contact. (b) Point-contact (PC) resistance, R_{PC} as a function of the change in effective PC diameter, Δd_{PC} .

2. Estimation of the inelastic scattering length

We have performed the point contact measurements at $T \sim 6$ K, where electron-phonon scattering is significantly suppressed since the Debye temperature in these films is ~ 170 K [shown in Extended Data Fig. 6 of main manuscript]. This is established by the saturation of $\rho(T)$ at these temperatures.[See Extended Data Fig. 3 of main manuscript]. To have a quantitative estimation of $l_{in} \sim l_{e-ph}$, let us look at the electron-phonon relaxation length that can be represented as: $l_{e-ph} = v_F/\tau_{e-ph}$, where $\tau_{e-ph}^{-1} = (2\pi k_B/\hbar)\lambda T$. For λ ranging from $0.2 - 20$ that is obtained by fitting the $\rho - T$ data, this varies from $10^{-7} - 10^{-9}$ nm, shown by red-colored points in Fig. 3(b). However, this is an overestimation of l_{in} since the scattering rate varies superlinearly, $\rho \sim T^5$ at lower temperatures, as can be understood from the asymptotic limit of the Bloch-Grüneisen expression of resistivity from electron-phonon scattering (discussed in Methods as well Supplementary Information). We have performed quantum transport measurements at $T \sim 7$ K to extract l_{in} . Fig. 3(a) shows the variation of the transverse resistance of a typical film with $F \approx 0.12$ in perpendicular magnetic fields (B) within $B \pm 5$ T. By fitting the data with the three-dimensional analogues of the Hikami-Larkin-Nagaoka expressions of weak localization/anti-localization [6], we extract the phase coherence length, l_ϕ . This is equivalent to l_{in} under the assumption that there are no magnetic impurities in the system. The estimated l_ϕ is shown by blue-coloured points in Fig. 3(b). Further details on the quantum transport measurements will be shown in another manuscript (*Kumbhakar et.al*).

Figure 3. **Inelastic scattering length:** (a) Magnetotransport in a film with $F = 0.12$ within perpendicular magnetic fields of $B \pm 5$ T. (b) Comparison of the inelastic scattering length (l_{in}) computed in two different ways. Error bars in the red-coloured points have been computed from the error in resistivity, ρ , as shown in Fig. 1g of the main manuscript by error propagation. Error bars in the blue-coloured points represent the standard deviation of the channel-to-channel statistics of l_{ϕ} , estimated from the fits to the quantum transport data of the particular channel.

3. Transport regime of point contact

Fig. 4 shows the variation of l_{el} , l_{in} , Λ , and d_{PC} with Ag-filling fraction F at $T \sim 6 - 8$ K. d_{PC} shown in the figure corresponds to the dimension of the point contact orifice for the spectrum, shown in Fig. 2c of the main manuscript. l_{in} has been derived from the quantum transport measurements as described above. We observe $l_{\text{el}}, \Lambda \ll d_{\text{PC}} \leq l_{\text{in}}$, indicating that the point contact is at the crossover between the diffusive and the thermal regimes, which we call the *quasi-thermal* regime. We wish to highlight that l_{in} is the lower limit of l_{e-ph} since we did not eliminate the contribution of the inelastic processes from electron-electron scattering. This would mean that the point contact lies closer to the diffusive regime than indicated in Fig. 4. **Hence, we have considered the value of $f \sim 1.1$ in the estimation of λ .**

Figure 4. Comparison of elastic (l_{el}), inelastic (l_{in}), electron diffusion ($\Lambda = \sqrt{l_{\text{el}} l_{\text{in}} / 3}$) length scales, and the point contact diameter (d_{PC}) for different values of Ag-filling F at temperature $T \sim 6 - 8$ K. For $F \gtrsim 0.3$, l_{el} becomes lesser than the interatomic spacing, indicating the inapplicability of the Drude expression at these resistivities to estimate the elastic scattering length. Error bars in l_{el} have been computed from the error in resistivity, ρ , as shown in Fig. 1g of the main manuscript by error propagation. l_{in} shown here corresponds to the phase breaking length, l_{ϕ} estimated from quantum transport measurements, shown in Fig. 3. Error bars in Λ have been computed from the errors in l_{el} , and l_{in} by error propagation.

4. Consistency of the analysis

We present here the normalized point contact spectrum, $\rho/R_{\text{PC}} dR_{\text{PC}}/dV$ measured at different values of zero bias $R_{\text{PC},0}$ at different positions at temperatures $T \sim 6 - 8$ K. The dotted line is a guide to the eye indicating the value of the normalized spectra at $5\hbar\omega_D \sim 90$ meV, $\omega_D \approx 18$ meV being the Debye frequency of the films. As from where λ has been estimated.

We observe the point contact spectrum scales at different values of $R_{\text{PC},0}$, indicating

1. the accurate estimation of the PC diameter, and
 2. the quantity $\rho/d_{\text{PC}} dR_{\text{PC}}/dV$ to be corresponding to a physical entity intrinsic to the system, supporting the Eq. 20
- For the film with $F = 0.33$, we observe the prominence of the peaks for certain values of point contact resistance. However, the background is equal in both cases, supporting the consistency of the background.

Figure 5. Normalized point contact spectrum ($\rho/R_{\text{PC}} dR_{\text{PC}}/dV$) measured at different positions with different zero bias point contact resistances ($R_{\text{PC},0}$) for films with Ag-filling fraction $F = 0.33, 0.5, 0.7$ at temperatures $T \sim 6 - 8$ K.

3. Comment on any other T -dependent scattering mechanisms: We observe from Fig. 2c of the main manuscript and Fig. 5 that the background value of the PCS spectrum starts saturating at energy scales of ~ 20 meV, which corresponds to the acoustic phonon modes in Au. This energy scale is independent of the Ag fraction and is spatially homogeneous, as can be inferred from the scaling of the normalized point contact spectrum at different positions. These observations exclude any disorder-induced inelastic scattering that is generally stochastic in nature. We also exclude the possibility of electron-electron interaction to the dominant inelastic scatterer in the experimentally measured region of bias ± 100 meV since electron-electron interaction in metals is generally insignificant at these temperatures ($T \sim 6 - 8$ K).

-
- [1] Naidyuk, Y.; Yanson, I. *Point-Contact Spectroscopy*; Springer Series in Solid-State Sciences; Springer New York, 2019.
 - [2] Van Gelder, A. On the structure of the d^2J/dV^2 characteristics of point contacts between metals. *Solid State Communications* **1980**, *35*, 19–21.
 - [3] Kulik, I. Frequency dispersion caused in the conductivity of metal microcontacts by nonequilibrium-phonon relaxation. *Soviet Journal of Experimental and Theoretical Physics Letters* **1985**, *41*, 370.
 - [4] Kulik, I. O.; Ellialtiogamalu, R. *Quantum mesoscopic phenomena and mesoscopic devices in microelectronics*; Springer Science & Business Media, 2012; Vol. 559.
 - [5] Kulik, I. On the determination of $2F()$ in metals by measuring I–V characteristics of “wide” (non-ballistic) point-contact junctions. *Physics Letters A* **1984**, *106*, 187–190.
 - [6] Baxter, D. V.; Richter, R.; Trudeau, M.; Cochrane, R.; Strom-Olsen, J. Fitting to magnetoresistance under weak localization in three dimensions. *Journal de Physique* **1989**, *50*, 1673–1688.

Response to Reviewers

Comments from Reviewer-1: NCOMMS-24-44901A

In the revised manuscript "Engineering ultra-strong electron-phonon coupling and nonclassical electron transport in crystalline gold with nanoscale interfaces " by Kumbhakar et al., the authors addressed my main concern about the point contact regime. It is the thermal one in their case and not the diffusive one as originally claimed. In the new version they properly show that while the spectroscopic information on the electron-phonon function is lost the general information on the electron-phonon coupling constant can be estimated also in the thermal regime. The detailed explanation is provided in the Supplementary information, Section V. With this I suggest to publish the paper in the Nature Communications.

We thank the reviewer for the comment.

There are errors/misprints in the Supplementary Information Section V., which most probably do not spoil the overall analysis but should be corrected:

In

A. Electron-phonon coupling ...

1. Ballistic regime

- Diameter of the PC is denoted once as d_{PC} but next as d

We have rectified this in Eq. [S8] of Supplementary Information and consistently denoted the point contact diameter as d_{PC} everywhere.

In Wexler formula (S8) Γ is equal to unity at $l_{el} \ll d_{PC}$ which is the diffusive NOT THE BALLISTIC regime.

We have rectified this and mentioned correctly that the limit $l_{el} \ll d_{PC}$ is the diffusive regime.

Eq. S10 is reversed but then correctly used in Eq. S24.

We have rectified Eq. [S10].

Summary after 3. Thermal regime, first bullet wrongly reads Eq. 11a instead of Eq. S19a

B. Analysis of experimental data.

We have rectified this and correctly referred to Eq. [S19].

Typical residual resistivities should be some 10^{-6} Ohm.m AND NOT 10^{-6} microOhm.m as stated

- the estimated mean free path (0.1 nm) can hardly be larger than the typical PC diameter

We have rectified this and correctly mentioned the resistivity as 10^{-6} Ω . m.

l_{e-ph} is defined as the Fermi velocity divided by the e-ph relaxation time?

To estimate, l_{e-ph} we have carried quantum transport measurements that lead us to the phase breaking length, which is equivalent to the inelastic scattering length.